Wild plant species growing closely connected in a subalpine meadow host distinct root-associated bacterial communities

Aleklett Kristin 1
Leff Jonathan W. 2
Fierer Noah 2
Hart Miranda 1 miranda.hart@ubc.ca
1 Department of Biology, University of British Columbia—Okanagan , Canada
2 Cooperative Institute for Research in Environmental Sciences and the Department of Ecology and Evolutionary Biology, University of Colorado , Boulder, CO , USA
Smidt Hauke
Electronic publication date: 2015 Feb 26
Publication date: 2015
Volume: 3
Electronic Location ID: e804
Received 2014 Oct 20; Accepted 2015 Feb 6
Copyright: © 2015 Aleklett et al.
Copyright year: 2015
Copyright holder: Aleklett et al.
License: This is an open access article distributed under the terms of the Creative Commons Attribution License, which permits unrestricted use, distribution, reproduction and adaptation in any medium and for any purpose provided that it is properly attributed. For attribution, the original author(s), title, publication source (PeerJ) and either DOI or URL of the article must be cited.
License URL: https://creativecommons.org/licenses/by/4.0/

Keywords: Root microbiota, Host effect, Bacteria, Plant identity, Pilosella aurantiaca, Rhizosphere

Funding: NSERC Discovery Grant UBC Okanagan Internal Research Grant Program F12-00704 This project was funded through a NSERC Discovery Grant to Miranda Hart, and the support of UBC Okanagan Internal Research Grant Program (F12-00704) to Kristin Aleklett. The funders had no role in study design, data collection and analysis, decision to publish, or preparation of the manuscript.

==============================
Plant roots are known to harbor large and diverse communities of bacteria. It has been suggested that plant identity can structure these root-associated communities, but few studies have specifically assessed how the composition of root microbiota varies within and between plant species growing under natural conditions. We assessed the community composition of endophytic and epiphytic bacteria through high throughput sequencing using 16S rDNA derived from root tissues collected from a population of a wild, clonal plant (Orange hawkweed–Pilosella aurantiaca) as well as two neighboring plant species (Oxeye daisy–Leucanthemum vulgare and Alsike clover–Trifolium hybridum). Our first goal was to determine if plant species growing in close proximity, under similar environmental conditions, still hosted unique root microbiota. Our results showed that plants of different species host distinct bacterial communities in their roots. In terms of community composition, Betaproteobacteria (especially the family Oxalobacteraceae) were found to dominate in the root microbiota of L. vulgare and T. hybridum samples, whereas the root microbiota of P. aurantiaca had a more heterogeneous distribution of bacterial abundances where Gammaproteobacteria and Acidobacteria occupied a larger portion of the community. We also explored the extent of individual variance within each plant species investigated, and found that in the plant species thought to have the least genetic variance among individuals (P. aurantiaca) still hosted just as diverse microbial communities. Whether all plant species host their own distinct root microbiota and plants more closely related to each other share more similar bacterial communities still remains to be fully explored, but among the plants examined in this experiment there was no trend that the two species belonging to the same family shared more similarities in terms of bacterial community composition.

Introduction

Plant roots function as distinct habitats within the soil and bacterial communities in root systems have repeatedly been shown to differ from those of the surrounding bulk soil (Smalla et al., 2001; Haichar et al., 2008; Gottel et al., 2011; Lundberg et al., 2012). Even though root associated bacterial communities (both rhizospheric—in the soil surrounding the roots, epiphytic—living at the surface of roots and endophytic—living inside root tissues) have been under investigation for many years, there is still little consensus in how these communities are formed and what determines their composition (Berg & Smalla, 2009; Aleklett & Hart, 2013; Bulgarelli et al., 2013).

Traditionally, the composition of bacterial communities living in association with plants has been attributed to environmental factors. For example, soil type has been suggested as the strongest determinant of community structure in root associated microbial communities (de Ridder-Duine et al., 2005; Singh et al., 2007; Lundberg et al., 2012; Bulgarelli et al., 2013). At the same time, it has also been argued that the host plant may play an equally large role in determining the composition of its root microbiota (Marschner, Grierson & Rengel, 2005; Costa et al., 2006; Hartmann et al., 2008; Doornbos, Van Loon & Bakker, 2011), especially endophytic bacterial communities (Haichar et al., 2008).

Recent work has demonstrated that hosts can alter their root microbiota by regulating soil conditions in the vicinity of the root system through root exudation of sugars, phenolics and amino acids that could also function as signaling molecules with the microbes in the surrounding soil (Chaparro et al., 2013). Since root exudation patterns and composition can be associated with plant gene expression, variation in host genetics has the potential to create large differences in the chemical profile of plants and consequently the composition of microbes able to inhabit the root system. Several studies have found that different plant species or genotypes of the same species host distinct microbial communities (Bailey et al., 2005; Marschner, Grierson & Rengel, 2005; Van Overbeek & Van Elsas, 2008; Schweitzer et al., 2008; Micallef et al., 2009; Manter et al., 2010; Becklin, Hertweck & Jumpponen, 2012; Peiffer et al., 2013). Even studies where soil type was considered to have the strongest effect on structuring the root microbiota, differences in bacterial community composition between genotypes was still detected (Bulgarelli et al., 2012; Lundberg et al., 2012).

The root environment varies greatly among plant species (Bardgett, Mommer & De Vries, 2014); these differences may lead to the selection of distinct bacterial communities. Plants can differ in terms of both root lifespan (Roumet, Urcelay & Díaz, 2006), root architecture (Hodge et al., 2009), root surface structure and components and patterns of root exudation (Bais et al., 2006). Root exudates are known to provide a food source for the microbes (Farrar et al., 2003), instigators of symbiotic associations (such as mycorrhizal infection or nodule formation) (Bais et al., 2004), and defend the plant against pathogens (Doornbos, Van Loon & Bakker, 2011). All these plant characteristics could contribute to shaping the root systems of different plants into local habitats and potentially distinct niches for microbial colonizers.

The role of intra-species variation among root associated microbial communities has been overlooked, but might represent a significant proportion of variation in natural systems (Bell et al., 2014). Since we know that natural populations exhibit variation in root exudation patterns and root morphology, one would expect there to be variation among individual plants in their root microbiota as well (Micallef et al., 2009). But variation among plants might also be driven by environmental heterogeneity because we know that small-scale environmental heterogeneity exists in soil systems. Is this variation static across plant taxa, or do different taxa exhibit more variation than others? If plant genetics are determining bacterial community composition, then certainly the populations with low genetic diversity (i.e., asexually reproducing, or metapopulations) would be expected to have less variation than sexually reproducing populations with high levels of gene flow. Because we sampled a plant species known to reproduce clonally through stolons and apomixis (P. aurantiaca), we also examined whether individuals within that species had less dispersion in their microbial community composition than individuals within the other two plant species.

The majority of studies characterizing bacterial communities in the root microbiota have been conducted with model plants in artificial greenhouse settings or agricultural contexts (Marschner & Yang, 2001; Garbeva, Van Veen & Van Elsas, 2004; Micallef et al., 2009; Manter et al., 2010; Doornbos, Van Loon & Bakker, 2011; Lundberg et al., 2012) where the study of genetically modified plants have been especially informing when it comes to understanding slight differences between plant genotypes (Van Overbeek & Van Elsas, 2008; Weinert et al., 2009; İnceoğlu et al., 2010). While these studies are crucial for understanding the mechanistic basis of plant:microbe interactions, they do not reflect how natural environmental conditions contribute to variation in bacterial community composition across individual plants, particularly in complex environments where a wide diversity of plants and biota are interacting.

In this study, we explored variation in bacterial community composition between individual root systems of neighboring plants in a common field in order to determine how much variation exists within and between plant taxa. We sampled the root microbiota of three plant species growing within 10 m from each other in a field and asked—are bacterial root communities distinct among plant species growing in a common location, close enough to be exposed to the same environmental conditions? And—do certain plant species contain more intra-species variance in bacterial communities than others?

Methods

Field site and target plant

Field site description

Samples were collected in August, 2011, from a subalpine meadow near Chute Lake, British Columbia, Canada (49.698859N, −119.533133W). The sampling area has not been used for agriculture or forestry but is in proximity to a forestry road as well as a camp site. Since it contains a high number of invasive plant species, it could also be considered a disturbed site. The soil at the site was determined to be a sandy loam, and the site is classified under the biogeoclimatic zone Interior Douglas Fir, dry warm (IDFdw) (Biogeoclimatic Ecosystem Classification (BEC) and Ecology Research program of the British Columbia). This zone is characterized by cold winters and dry, warm summers with low amounts of precipitation due to the rain shadow created by the Coast, Cascade and Columbia mountains. Because the climate is very dry in the summers, forest fires are a common feature in the landscape, and the site is located close to the area that was badly burnt during a large fire in Okanagan Mountain Park in 2003. The plant community sampled for this experiment was growing up on a rock plateau, away from the tree coverage, completely exposed to the elements (see Fig. 1 in additional material). The dominant vegetation at the field site consisted of Orange hawkweed (Pilosella aurantiaca (L.) FW Schultz and Schultz-Bip), Hairy vetch (Vicia villosa, Roth), Oxeye daisy (Leucanthemum vulgare, Lam.), Wild strawberry (Fragaria virginiana, Duchesne) Timothy (Phleum pratense, L.) and Alsike clover (Trifolium hybridum, L.) growing homogenously, but at different abundances across the field.

Target plant

Our target plant was P. aurantiaca (formerly known as Hieracium aurantiacum), which is native to Europe and invasive in North America. Because of P. aurantiaca’s stoloniferous reproduction, the species is able to form dense mats in the landscape, making it hard for other plant species to compete for space through seed germination in the field (Giroday & Baker, 2006). This is one of the features that have made the species a successful invasive plant within British Columbia (Giroday & Baker, 2006). Genetic diversity within P. aurantiaca has previously been examined across 48 locations in North America, and results showed that there were only three genotypes, of which two were found only in isolated locations (one in Alaska and one in Oregon) (Loomis & Fishman, 2009). By choosing to work with a plant expressing this low diversity in wild populations, we hoped to minimize genetic variance within the population that we sampled.

To clarify the role of host identity and intra-species variance in bacterial root microbiota, we additionally sampled two of the co-occurring plant species, L. vulgare and T. hybridum that were in the same developmental stage (flowering) as P. aurantiaca.

Experimental design

Root systems of P. aurantiaca were collected one meter apart along two 10 m transects (n = 20) in order to explore within species variance of a clonal plant. To be able to compare bacterial community composition between species, additional samples of T. hybridum (n = 10) and L. vulgare (n = 10) were also collected where present along the transects, several of which were growing within centimeters of P. aurantiaca samples. Because T. hybridum and L. vulgare were not as abundant within the field, their original sample size was lower. For statistical comparisons between species however, an equal sample size of 8 plants/species was used throughout analyses.

Each root system was rinsed from surrounding rhizospheric soil in de-ionized water in order to separate it from roots of neighboring plants. Root systems were then cut up in pieces and a subsample of root tissue, representative of the whole root system, including young fresh roots as well as older root tissues (with no exclusion of nodules in T. hybridum), was collected and further used for classification of bacterial community composition. Since no further treatment was performed in order to remove rhizoplane microbes, we assume that the communities extracted could be of either endophytic or epiphytic origin.

Bacterial community analysis

Amplification and sequencing of target gene

DNA from all collected plant tissues (0.25 g/sample) was extracted using a PowerSoil DNA Isolation Kit (MoBio Laboratories Inc., Carlsbad, California, USA) according to the manufacturer’s protocol. Microbial diversity and the relative abundances of individual taxa were assessed by barcoded pyrosequencing of a portion of the 16S rRNA gene. Each DNA sample was amplified in triplicate through PCR reactions using the protocol described in Fierer et al. (2008) except with a different primer pair. The forward primer contained the 454 Life Sciences primer B sequence, the bacterial primer 799f (Chelius & Triplett, 2001) and a two-base linker sequence (‘AG’). The reverse primer contained the 454 Life Sciences primer A sequence, a unique 12 bp error-correcting Golay barcode (Fierer et al., 2008), a ‘GT’ linker sequence, and the ‘universal’ bacterial primer 1115r (Reysenbach & Pace, 1995). The targeted gene region has shown to be appropriate for accurate taxonomic classification of bacterial sequences and the primers are designed to exclude chloroplasts from plant tissues in the samples (Redford et al., 2010). Amplicons were visualized via gel electrophoresis purified and quantified. Amplicons from all samples were then combined in equimolar ratios into a single tube. Samples were sequenced at Engencore (University of South Carolina) on a Roche GS-FLX sequencer running the Titanium chemistry.

Processing raw sequence data

All sequences were de-multiplexed and further analyzed using the Quantitative Insights Into Microbial Ecology (QIIME) toolkit (Caporaso et al., 2010). Operational Taxonomic Units (OTUs) were defined at the level of ≥97% similarity and the taxonomy assigned by comparing a representative sequence from each OTU to the Greengenes database (February 14th 2012 version) (DeSantis et al., 2006) using the Basic Local Alignment Search Tool (BLAST) classifier (Altschul et al., 1990). In order to correct for differences in the number of sequences analyzed per sample, a randomly selected subset of 400 sequences per root sample was used to compare relative differences in taxonomic diversity. Only samples from which we obtained a minimum of 400 bacterial sequences per sample or more were considered in the study, eliminating 3 samples from the study (one P. aurantiaca and two L. vulgare). Though 400 sequences cannot fully capture the rare biosphere, it allowed us to compare samples while still maintaining as many samples as possible. It has previously been shown that studies of bacterial communities show similar results even at a lower rarefaction (Hamady & Knight, 2009; Kuczynski et al., 2010). In fact, re-analyzing our data set with a higher rarefaction limit showed the same general trends but drastically lowered our number of samples available to analyze.

Statistics

Differences in community composition between samples were calculated using phylogenetic metric (UniFrac) where weighted UniFrac shows an emphasis on the more abundant taxa in samples and un-weighted UniFrac treats all taxa the same (Lozupone et al., 2007; Hamady, Lozupone & Knight, 2010). As a comparison, we also included a taxonomic metric (Bray-Curtis distance) to explore whether dissimilarity patterns were the same in terms of presence/absence of taxa. Before calculating Bray-Curtis distances, all relative abundances were log-transformed. 2-D scatterplots of Principal Coordinates Analysis (PCoA) generated in PRIMER-E (Clarke & Gorley, 2006) were used to visualize the greatest amount of variability in the pair-wise distances between samples.

We tested for variance among host plants in their root microbiota using a 2-way PERMANOVA (Anderson, 2005) with host species and transect as factors and weighted and un-weighted UniFrac as well as Bray Curtis as our dissimilarity metrics. All analyses were permutated 9999 times. Since it has been shown that PERMANOVA is not robust when sampling efforts are un-equal (Anderson & Walsh, 2013), we subsampled 8 samples from each species which were used for both PERMANOVA and PermDISP analyses.

Variability in community composition within each of the three species was analyzed through PermDISP (Anderson, 2004) (9,999 permutations), creating a centroid for each species and measuring the average spread of samples belonging to that species from the centroid. A large spread (high average) would indicate a high variability in community composition among individuals within the species (Anderson, 2004).

Results

Variation between host species

When comparing the phylogenetic overlap between bacterial root microbiota (UniFrac) across three different species of plant hosts growing in a common field, bacterial communities from samples of the same plant species were significantly more similar to each other than to bacterial communities sampled from plants of the two other species (Table 1). This was true for both weighted (Pseudo-F = 8.54 p = 0.0001) and un-weighted UniFrac (Pseudo-F = 1.66 p = 0.0001) as well as Bray Curtis dissimilarities (Pseudo-F = 2.27 p = 0.0001) (Table 1). These patterns were also evident from the principal coordinates analyses which showed little overlap between samples of different plant species (Fig. 1).

Figure 1 Principal coordinates analysis plot illustrating the phylogenetic overlap in root prokaryotic community composition among samples from three different plant species.

Phylogenetic overlap between communities was assessed using weighted UniFrac. Community composition was significantly different among plant species (P < 0.001; PERMANOVA).

Table 1 PERMANOVA results.

Comparing bacterial community resemblance between plant species and transects using different diversity metrics.

Factor	Diversity metric	Pseudo-F	P (perm)	
Species	Weighted UniFrac	6.15	0.0001	
	Un-weighted UniFrac	1.43	0.0001	
	Bray Curtis	1.88	0.0001	
Transect	Weighted UniFrac	1.22	0.23	
	Un-weighted UniFrac	1.06	0.22	
	Bray Curtis	1.07	0.24	
SpXTr	Weighted UniFrac	1.54	0.09	
	Un-weighted UniFrac	1.03	0.28	
	Bray Curtis	1.06	0.20	

Variation within host species

Plant species differed in how much variance there was among bacterial communities of individual root samples (Fig. 1). There was a significant difference between plant species in the amount of compositional dissimilarity of bacterial taxa between individual plants (Bray Curtis: F = 9.56 p = 0.02). That is, the amount of dispersion of individual plants from the centroid differed among plant species. In this case, P. aurantiaca exhibited the most variance among plant individuals, while T. hybridum showed the least (Table 2). This was not the case when the same data was analyzed using phylogenetic measures, since we could not detect any significant difference in dispersion among plant taxa (UniFrac: weighted F = 2.57 p = 0.15; un-weighted F = 2.56 p = 0.63) (Table 2).

Table 2 PermDISP results showing the average spread from centroid and standard error (SE) for samples of each species.

The PERMANOVA (P(perm)) values are assessing whether there is a significant difference between species in sample dispersion, using different diversity metrics.

Diversity metric	F	P (perm)	Species	Average	SE	
Weighted UniFrac	3.10	0.11	P. aurantiaca	9.96 E−2	4.28 E−3	
			T. hybridum	8.57 E−2	5.70 E−3	
			L. vulgare	0.12	1.29 E−2	
Un-weighted UniFrac	1.04	0.37	P. aurantiaca	0.47	3.39 E−3	
			T. hybridum	0.46	5.42 E−3	
			L. vulgare	0.47	7.25 E−3	
Bray Curtis	9.30	0.005	P. aurantiaca	51.43	0.73	
			T. hybridum	47.13	0.44	
			L. vulgare	51.05	1.05	

Relative abundance of taxa across hosts

A total number of 4,384 unique OTUs were analyzed within the rarefied data set. A taxonomic summary, showing the average abundance of bacterial phyla in P. aurantiaca, T. hybridum and L. vulgare samples, illustrates the compositional differences between root systems of different plant species (Fig. 2). In P. aurantiaca, the most abundant phylum was Betaproteobacteria which made up, on average, 29% out of all sequences found in P. aurantiaca samples, followed by Bacteroidetes (19%), Alphaproteobacteria (16%) and Actinobacteria (12%). In T. hybridum, Betaproteobacteria made up, on average, 51% of the all bacterial sequences found in the species, followed by Alphaproteobacteria (21%) and Bacteroidetes (16%). Bacterial communities in L. vulgare samples were, similarly to T. hybridum, dominated by Betaproteobacteria (50%), followed by Bacteroidetes (18%) and Alphaproteobacteria (12%). A table showing the relative distribution of the 14 most abundant OTUs across all samples as well as their relative abundance within samples of the different species is given in Table 3.

Figure 2 Comparison of the average bacterial community composition and relative abundances, at the phylum level (Proteobacteria divided into class) in root samples from three different plant species.

Results show a strong dominance of sequences belonging to Betaproteobacteria in all three plant species, but especially in T. hybridum (51%) and L. vulgare (50%). The phyla representing less than 1% out of the total community were grouped as “Other” and consisted of: NKB19, Nitrospirae, PAUC34f, Cyanobacteria, Elusimicrobia, Fibrobacteres, Chlamydiae, SC4, Spirochaetes and Thermi. Sequences not matching the database were recorded as “No blast hit.”

Table 3 The core root microbiota represented by the fourteen OTUs with the highest abundances across all samples.

Values are calculated as the average percent out of the 400 sequences recorded for each sample, across all species (Total %). As a comparison, data is also included for what percentage (on average) the fourteen OTUs make up within the bacterial communities of the three plant species examined (P. aurantiaca, T. hybridum and L. vulgare).

# OTU
ID	Total
%	P.
aurantiaca
%	T.
hybridum
%	L.
vulgare
%	Phylum	Class	Order	Family	Genus	Species	
1,537	7.6	2.8	9.4	16.8	Proteo-
bacteria	Beta-
proteobacteria	Burk-
holderiales	Oxalo-
bacteraceae	Herba-
spirillum		
19,032	3.1	3.2	3.9	2.2	Proteo-
bacteria	Beta-
proteobacteria	Burk-
holderiales				
22,328	2.7	1.0	5.1	3.8	Proteo-
bacteria	Beta-
proteobacteria	Burk-
holderiales	Coma-
monadaceae	Limno-
habitans		
30,435	1.5	1.7	1.3	1.2	Proteo-
bacteria	Beta-
proteobacteria	Burk-
holderiales		Methy-
libium	Methylibium
petroleiphilum	
4,453	1.3	0.8	1.7	2.1	Proteo-
bacteria	Beta-
proteobacteria	Burk-
holderiales	Oxalo-
bacteraceae	Janthino-
bacterium	Janthinobacterium
lividum	
1,231	1.2	1.9	0.5	0.4	Bacteroi-
detes	Sphingo-
bacteria	Sphingo-
bacteriales	Flexi-
bacteraceae	Cytophaga		
22,285	1.0	1.5	1.0	0.4	Proteo-
bacteria	Alpha-
proteobacteria	Rhizobiales	Brady-
rhizobiaceae	Brady-
rhizobium		
20,009	1.0	1.5	0.2	0.6	Actino-
bacteria	Actino-
bacteria	Actino-
mycetales	Thermomon-
osporaceae	Actino-
corallia	Actinocorallia
longicatena	
25,072	0.9	1.0	1.0	0.7	Proteo-
bacteria	Gamma-
proteobacteria	Xantho-
monadales	Xantho-
monadaceae	Rhodano-
bacter	Rhodanobacter
lindaniclasticus	
1,340	0.9	1.0	1.0	0.6	Bacteroi-
detes	Sphingo-
bacteria	Sphingo-
bacteriales	Sphingo-
bacteriaceae	Sphingo-
bacterium	Sphingobacterium
faecium	
5,213	0.8	0.6	0.9	1.3	Bacteroi-
detes	Sphingo-
bacteria	Sphingo-
bacteriales	Sphingo-
bacteriaceae			
29,492	0.8	1.3	0.1	0.6	Proteo-
bacteria	Gamma-
proteobacteria	Chro-
matiales	Sino-
bacteraceae			
26,917	0.8	1.1	0.3	0.7	Chloroflexi	Chloroflexi	Roseiflexales	Kouleo-
thrixaceae	Kouleo-
thrix		
1,184	0.8	0.4	1.3	1.1	Bacteroi-
detes	Flavo-
bacteria	Flavo-
bacteriales	Flavo-
bacteriaceae	Chryseo-
bacterium		

A closer examination of the relative abundances of all Betaproteobacteria found in samples showed that while Burkholderiales was the predominant order across all three species, T. hybridum and L. vulgare samples were heavily dominated by bacteria of the family Oxalobacteriaceae—especially bacteria of the genus Herbaspirillum (11% of the total bacterial community in T. hybridum and 18% in L. vulgare) (Fig. 3).

Figure 3 Average relative abundances of Betaproteobacteria families and Oxalobacteraceae genera found in root samples of the three plant species.

Values are given as the percentage of sequences belonging to a certain taxa out of the total average bacterial community for each of the three plant species (rarefied at 400 sequences/sample). The heat map is colour coded from blue (low abundance) to red (high abundance).

Discussion

Host specificity

Our study shows that root bacterial communities vary significantly between plants belonging to three different species, growing in close proximity to each other in natural plant communities. Because the plant roots collected in our study were growing in close proximity to each other, it is unlikely that variation in soil conditions (usually thought to be one of the main drivers of microbial community structure) is a significant source of variation among our samples.  While these results need to be further confirmed across multiple field sites in order to draw general conclusions about whether this is just a local trend or a general pattern across plant species and soil types, our results support previous work showing bacterial host plant specificity in roots of agricultural crops (Marschner & Yang, 2001; Wieland, Neumann & Backhaus, 2001; Haichar et al., 2008) and wild grass species (Kuske & Ticknor, 2002; Osanai et al., 2012).

Although all plant species investigated in this study (P. aurantiaca, T. hybridum and L. vulgare), are perennial, there are significant differences in root morphology between the species. For example, P. aurantiaca and L. vulgare (both belonging to the family Asteraceae) have creeping root stocks and produce fibrous root systems whereas T. hybridum (family Fabaceae) grows a branching tap root system that is known to form nodules with nitrogen fixing bacteria. This variation in root morphology could contribute to the differences in abundance and composition of bacteria in our results (Bardgett, Mommer & De Vries, 2014). For example, roots that penetrate deeper soil may encounter different microbes than those in shallow layers (Fierer, Schimel & Holden, 2003). Similarly, the thickness and/or texture of the root surface (i.e., woody, fibrous) may be more or less penetrable to colonizing bacteria.

Part of the variance seen in bacterial community composition between the three plant species could also be caused by species-specific root exudation patterns. For example, several members of the Asteraceae family are known to produce allelochemicals that could affect the bacterial community as well as surrounding plants (Alford, Vivanco & Paschke, 2009). However, these differences are difficult to assess in wild plant communities, especially when roots of different plant species grow in close proximity to each other with entangled root systems. In our study, the roots grew so intimately that exudation from one plant species could have influenced root systems of neighboring plants.

Individual variation in root microbiota

We know that genetic differences between plants, even at the genotype level, can affect the composition of the root microbiota (Bailey et al., 2005; Schweitzer et al., 2008; Peiffer et al., 2013). Thus, we would expect variation in bacterial community composition among individuals within a population of plants, even when they are growing in a common environment, due to genetic variation in the population in terms of root traits and exudation chemistry, among other factors. Though there is a potential for clonality in T. hybridum, we still predicted less individual variance in the root microbiota among individuals from P. aurantiaca (thought to consist of mainly one genotype across all of North America (Loomis & Fishman, 2009)) than within the two co-occurring out-crossing plant species with presumed higher genetic diversity (L. vulgare and T. hybridum).

Though our data show a significant difference in compositional turnover within different plant species, it rejects the hypothesis that P. aurantiaca had the most similar root communities across individuals. Comparing the average dispersion of bacterial community composition for the three plant species, there was no indication that P. aurantiaca had a smaller dispersion than the two other plant species (Table 2). Instead, it shows that P. aurantiaca had the highest variation within a species comparing dispersion based on taxonomic differences (Table 2). The fact that we could not detect any differences in dispersion when using phylogenetic metrics suggests that individual root systems differ more in terms of which taxa are present or absent than how related they are, or that there is little phylogenetic conservatism at the individual level.

Overall, this study shows that the extent of individual variation seen in root microbiota varies between species, but that a plant species thought to be more genetically homogenous does not necessarily host more homogeneous root communities. It also indicates that individual variation in bacterial community composition in root systems is determined, not only by plant genetics, but also by small scale variance in the surrounding environment (soil chemistry, humidity, biotic interactions etc.) and potentially, events throughout the plant’s life that could affect root colonization (Aleklett & Hart, 2013).

Bacterial community composition

Similar patterns of bacterial community composition to what we found in our plants, growing in a subalpine meadow in Canada, have been reported in rhizosphere samples of other studies. For example, roots tissues of the plant species that we sampled were mainly dominated by Betaproteobacteria, (Fig. 2), especially members of the order Burkholderiales and the family Oxalobacteriaceae, which represented as much as 32% of the total bacterial community in L. vulgare (Fig. 3). Seed- and root-colonizing populations of Oxalobacteriaceae have previously shown to be responsive to plant species (Green et al., 2007), supporting our data of plant species hosting distinct bacterial communities. Dominance by these taxa in root systems has also been reported in other studies. For example, roots of Arabidopsis thaliana, examined at the same taxonomic resolution by Lundberg et al. (2012), were dominated by Betaproteobacteria and Oxalobacteriaceae both in samples of rhizosphere soil as well as in the endophytic root compartment. In sphagnum mosses, Burkholderiales has also been documented as one of the dominant bacterial groups, thought to be behind the production of anti-fungal compounds and anti-microbial properties of the sphagnum mosses (Opelt et al., 2007).

Other studies have found Actinobacteria to dominate in root tissues of plants (e.g., Ottesen et al., 2013), especially in communities of the endophytic compartment (Bodenhausen, Horton & Bergelson, 2013). In our study, Actinobacteria represented at most 12% out of the total bacterial community in the plant species that we sampled (Fig. 2) and was mainly found in samples of P. aurantiaca that, in general, were less dominated by beta-proteobacteria.

The dominance of sequences belonging to the genus Herbaspirillum was further emphasized when we examined the fourteen most abundant OTUs across all samples (Table 3). Herbaspirillum spp. are known to colonize apoplastic or intracellular spaces of plant tissues and several species have shown the ablility to fix nitrogen (Schmid, Baldani & Hartmann, 2006). While it is believed that this nitrogen fixing ability could be beneficial to their plant host, it has also been documented that certain Herbaspirillum strains are mild pathogens and a causative agent of “mottled stripe disease” in crops such as sugar-cane (Schmid, Baldani & Hartmann, 2006). Besides Herbaspirillum, we also saw high abundances of sequences belonging to Limnohabitans and Cytophaga (Table 3), two genera more commonly associated with bacterial communities in fresh water (Kirchman, 2002; Simek et al., 2010) as well as the species Methylibium petroleiphilum, a recognized methylotroph (Kane et al., 2007) and Janthinobacterium lividum, known to thrive in soils (Shivaji et al., 1991) and produce antibiotics (Johnson, Tymiak & Bolgar, 1990). The high presence of these groups in our samples could be due to the inclusion of epiphytic members of the root microbiota, where bacteria associated with water films and soil particles of the root surface would be expected.

In comparison, Bodenhausen and colleagues (2013) found that a Flavobacterium of the phylum Bacteroidetes stood out as the single most abundant OTU in endophytic root samples, making up 10.15% of the total community. Though bacteria of the phylum Bacteroidetes represented a significant part of the community in root samples of the three plant species sampled in our study (Fig. 2), they were by no means the most dominant taxonomic group in any of the species (Table 3).

As the genus Trifolium are known to be hosts of nitrogen fixing bacteria that form nodules in their roots, T. hybridum samples were expected to host larger populations of Alphaproteobacteria, specifically belonging to the order Rhizobiales which is a common symbiont of legumes (Masson-Boivin et al., 2009). This expected pattern was not evident in our results though. The only OTU belonging to the order Rhizobiales detected in notable abudances in our study was a Bradyrhizobium taxa which made up 1% of the collective community of T. hybridum samples and 1.5% in P. aurantiaca samples (Table 3). Instead, it was evident that the T. hybridum community was dominated by the family Oxalobacteraceae (23.48%) (Fig. 3) and specifically one OTU of the genus Herbaspirillum (10.75%) (Table 3), which is mainly known to colonize roots of non-leguminous plants, and have nitrogen fixing properties (Baldani et al., 1997). What stands out though is that this group of bacteria was even more predominant in L. vulgare samples, where Oxalobacteraceae made up 32.47% of the community and the same Herbaspirillum OTU represented 18.26% of the total community.

Variance in relative abundances of bacterial taxa across plant species

We observed differences in the evenness of bacterial taxa across host plants. While the roots of L. vulgare and T. hybridum seemed dominated by a few select groups of microbes, samples of P. aurantiaca supported communities with abundances more evenly distributed among bacterial taxa (Fig. 2; Table 3). Though few studies have looked specifically at variance in bacterial evenness between plant species, it could be an important source of variation. For example, dominance of single taxon may indicate specialized plant/bacterial associations whereas high evenness in community composition could reflect generalist associations among plants and bacteria. Alternatively, differences in evenness may result from microbial interactions within the plant, not driven by the plant but microbial competition for plant resources.

Conclusions

In this study, we showed that plant identity plays a major role in explaining the variation seen in root microbiota both between and within plant species growing under natural conditions. Further studies across a larger set of wild plant species and natural sites as well as more detailed investigations of the effect of plant genetics versus plant phenotypic traits on bacterial community assembly could help resolve the relative contribution of host identity at an individual level in shaping the root microbiota. It would also allow us to draw further conclusions as to whether plats more related to each other actually host more similar bacterial communities across plant species and families.

The results of our study speak of how intimately related bacterial communities are with their host plants. Root systems of wild plants are never alone; they are constantly surrounded by the roots of other plants, entangled in the soil, competing for resources and space. Yet, our results show that bacterial communities associated with roots of plants growing in a common field are distinct between plant species.

Ultimately, we are not able to tell exactly why these three plant species have such distinct bacterial root communities, but further studies linking metabolomics of wild plants with bacterial community composition would be useful for better understanding how plants affect bacterial community assembly.

We would like to thank Monika Gorzelak for helping out with sampling efforts in the field.

Additional Information and Declarations

Competing Interests

Author Contributions

DNA Deposition

The authors declare there are no competing interests.

Kristin Aleklett conceived and designed the experiments, performed the experiments, analyzed the data, wrote the paper, prepared figures and/or tables, reviewed drafts of the paper.

Jonathan W. Leff and Noah Fierer contributed reagents/materials/analysis tools, reviewed drafts of the paper.

Miranda Hart conceived and designed the experiments, contributed reagents/materials/analysis tools, wrote the paper, reviewed drafts of the paper.

The following information was supplied regarding the deposition of DNA sequences:

FigShare DOI 10.6084/m9.figshare.1009707.

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
