# Peer review of "Wild plant species growing closely connected in a subalpine meadow host distinct root-associated bacterial communities"

_PeerJ, doi:10.7717/peerj.804_

## Round 0.1 · original submission · Major Revisions

As you can see from the individual reports, both reviewers agreed on the relevance and quality of your study.

Nevertheless, there are also some shortcomings that I would like you to address in a revised version, including the issues related to experimental design (number of samples, investigation of a single site, lack of soil physico-chemical data), as well as a number of suggestions for further improvement in style.

Reviewer 1 ·

Basic reporting

This is a well written manuscript, which fits to all points.

Experimental design

see point 2.(genral comments)

Validity of the findings

see point 2. (general comments)

Additional comments

The question if the soil microbiota or the plant species is the main driver of plant-associated microbial diversity is a long and conversely discussed question. Most of the published studies showed a clear effect of the plant species/cultivar but extend depends also on the phylogenetic position of the species. For example, medicinal plants show a stronger specificity than monocotyledonous plants (Köberl et al., Frontiers 2014). The authors analyze the microbial diversity of three wild plants from subalpine meadows, which is indeed less investigated than model plants and crops, and found distinct bacterial communities but with a different homogeneiity. Altogether this is a well-performed study but I have some comments to improve the quality.

1. The title is very poor, especially if you think on a beautiful subalpine meadow. Altogether more input from plant science would be of value; e.g., about the ecology of this vegetation community, plant species and physiology.
2. My main concern is that only one site was investigated. Are all these conclusions really statistically significant?
3. A novel aspect was the high extend of individual variation, which should be more discussed and mentioned in the abstract. Interestingly, I remember old studies which also showed these individual effects (based on cultivation) and came to the conclusion that is needs an appropriate statistical design and a lot of replicates.
4. There are interesting studies about mosses in natural bog ecosystems, which could be cited and discussed (Bragina, Sphagnum mosses).

Reviewer 2 ·

Basic reporting

This manuscript brings good insights on how plant species can host different microbial communities using 454 pyrosequencing technology. Data shown by the authors indicate that plants of different species host distinct bacterial communities in their roots.

Experimental design

There is a weakness in this study, as the authors did not include soil analyses (e.g.soil chemical/physical data) which could contribute to explain partially the findings of microbial biodiversity the rhizosphere of the three different plant hosts. The root environment should be taken into account, instead of only considering the host-microbes interaction. If these data can be added and discussed, this work will substantiate the findings in view of the comparisons made by the authors.

Also, it is unclear how many samples per plant species were used for the pyrosequencing and why they used 20, 10 and 10 for P. aurantiaca, T. hybridum and L. vulgare, respectively? Also, Figure 1 indicates that samples losses must have occurred for P. aurantica (1) and L. vulgare (2). Please, explain.

Validity of the findings

I have two main comments. First, the proposed hypothesis, specifically when based on the result of a single study and lack of mechanistic explanations. Second, there is only a single index of phylogenetic diversity. It seems that the authors should calculate multiple indices of biodiversity and see if they perform similarly, as 10 or 20 different samples were taken separately within the study area.

The Summary is quite poor and should contain more details of the methods used.

---

## Round 0.2 · accepted · Accept

You did an excellent job in improving your manuscript making use of the suggestions of the reviewers.

Reviewer 1 ·

Basic reporting

ok

Experimental design

ok

Validity of the findings

ok

Additional comments

ok